# Inhibition of IEC-6 Cell Proliferation and the Mechanism of Ulcerative Colitis in C57BL/6 Mice by Dandelion Root Polysaccharides

**DOI:** 10.3390/foods12203800

**Published:** 2023-10-17

**Authors:** Shengkun Yan, Lijun Yin, Rong Dong

**Affiliations:** 1School of Food Science and Nutrition Engineering, China Agricultural University, Beijing 100083, China; 2Agricultural Mechanization Institute, Xinjiang Academy of Agricultural Sciences, Urumqi 830091, China

**Keywords:** dandelion root polysaccharides, ulcerative colitis, Nrf2, ferroptosis, gut flora, metabolites

## Abstract

An exploration was conducted on the potential therapeutic properties of dandelion polysaccharide (DP) in addressing 3% dextran sodium sulfate (DSS)-induced ulcerative colitis (UC) in murine models. Subsequent assessments focused on DP’s influence on inflammation, oxidative stress, and ferroptosis in IEC-6 cells damaged by H_2_O_2_. Results highlighted the efficacy of DP in mitigating weight loss, improving disease activity index scores, normalizing colon length, and alleviating histological abnormalities in the affected mice. DP repaired colonic mitochondrial damage by enhancing iron transport and inhibited iron death in colonic cells. Moreover, DP played a pivotal role in enhancing the antioxidant potential. This was evident from the increased expression levels of Nrf2, HO-1, NQO-1, and GSH, coupled with a decrease in MDA and 4-HNE markers in the UC-afflicted mice. Concurrently, DP manifested inhibitory effects on MPO activation and transcription levels of inflammatory mediators such as IL-1β, IL-6, TNF-α, and iNOS. An upsurge in the expression of occludin and ZO-1 was also observed. Restoration of intestinal tightness resulted in decreased serum LPS and LDH levels. Thereafter, administration of DP by gavage increased fecal flora diversity and relative abundance of probiotics in UC mice. Analysis of metabolites indicated that DP counteracted metabolic disturbances and augmented the levels of short-chain fatty acids in ulcerative colitis-affected mice. In vitro studies underscored the role of DP in triggering Nrf2 activation, which in turn exhibited anti-inflammatory, antioxidant, and anti-ferroptotic properties. Summarily, DP’s capacity to activate Nrf2 contributes to the suppression of ferroptotic processes in intestinal epithelial cells of UC-affected mice, enhancing the intestinal barrier’s integrity. Beyond that, DP possesses the ability to modulate the gut microbiome, rectify metabolic imbalances, rejuvenate short-chain fatty acid levels, and bolster the intestinal barrier as a therapeutic approach to UC.

## 1. Introduction

Taraxacum officinale, also known as dandelion, is a member of the Asteraceae plant family and is a perennial herb native to the northern hemisphere [1,2]. There are 70 species in China, mainly distributed in the northwest and southwest, especially in Shanxi, Gansu, Qinghai, and Xinjiang are the most widely distributed, and their adaptability is strong to the natural world. This plant has long been used as a herbal medicine due to its antidiabetic, choleretic, antirheumatic, and diuretic properties [3]. Previous reports have demonstrated that almost all parts of this plant, such as flowers and leaves, have pharmacological activities [4,5,6]. However, several experiments have recently indicated an association between the chemical composition and bioactivity of dandelion root polysaccharides.

Ulcerative colitis (UC) represents a persistent, indeterminate IBD, predominantly impacting the colon and rectum. The origin of this condition remains elusive, with its manifestations restricted to either specific sections or the entirety of the colon and rectum [7]. Clinically, patients with UC are often associated with bloody stools, abdominal pain, low-grade fever, malnutrition, and even an elevated risk of colorectal cancer [8,9]. In recent years, the incidence and prevalence of UC have increased rapidly worldwide [10,11]. The initial goal of UC pharmacotherapy measures focuses on palliation of the disease, while the ultimate goal focuses on improving disability, preventing colectomy, and minimizing the risk of colorectal cancer [12]. Mild to moderate UC is widely treated with anti-inflammatory drugs such as 5-aminosalicylic acid and glucocorticoids [13,14], while severe UC is treated with immunosuppressants and biological therapies [15,16]. Declining quality of life has forced UC patients to seek diets with fewer side effects to prevent the development of UC, in which natural polysaccharides have shown advantages in preventing or intervening in intestinal disorders due to their low toxicity and wide range of bioactivities [17].

Earlier research indicates the potential of polysaccharides in alleviating colitis symptoms [18,19]. One hypothesis is that these compounds fortify the intestinal barrier while adjusting both the immune responses within the intestines and the overall microbial profile. Notably, they amplify the presence of beneficial microorganisms and suppress harmful ones [20]. Furthermore, botanical-derived polysaccharides appear to influence microbial metabolic pathways, with a keen interest in their impact on short-chain fatty acid synthesis [21]. Disruption in intestinal equilibrium, stemming from compromised epithelial barrier functions, plays a pivotal role in UC onset and advancement [22]. It’s worth noting that the inflammation [23] within intestinal epithelial cells, along with oxidative distress [23] and ferroptosis [24], ties back to the integrity of this intestinal barrier.

Polysaccharides, as one of the main active components in dandelion, have presented the functions of improving immunity, lowering blood lipids, preventing aging, and resisting radiation, and they have important potential applications in medicine and food [25]. By influencing fatty acid breakdown and countering microbial imbalances, dandelion derivatives alleviate the effects of colitis induced by dextran sodium sulfate [23]. However, it is not certain whether dandelion root polysaccharide (DP) exerts the anti-colitis effect, and the relevant molecular mechanism is not clear.

This research set out to gauge the remedial effects of DP on colitis using a DSS-treated mouse model. The intent was to elucidate the capacity of DP to mitigate colonic inflammation and shed light on the underlying mechanism. Through 16S rRNA sequencing, a thorough investigation into the variances and compositional structure of the colonic fecal microbiota was conducted. Subsequent metabolomic analyses aimed at identifying shifts in gut microbial metabolism, specifically focusing on fluctuations in short-chain fatty acids. Additionally, using the IEC-6 rat intestinal epithelial cell line, in vitro assessments were conducted to discern the anti-inflammatory, antioxidant, and anti-iron death properties of DP in the context of H_2_O_2_-induced inflammation.

## 2. Materials and Methods

### 2.1. Materials

Dandelion root was purchased from a local market in Hetian, Xinjiang, China in July 2022.

#### 2.1.1. Extraction, Isolation and Chemical Composition of DP

Using a methodology delineated in our prior research, we extracted DP from dandelion roots. The process began with thoroughly rinsing the roots with distilled water, followed by drying at 70 °C. Post-drying, the roots were pulverized, and the resulting powder underwent an extraction process with 95% ethanol to separate sedimentary components. Following a filtration step, the remaining solid was exposed to air until dry and subsequently subjected to a distilled water extraction. The resulting liquid was separated using centrifugation at 4000 rpm for a quarter hour, with this extraction step repeated two more times. Concentrating the pooled supernatants led to a reduction in volume to nearly a tenth of its initial volume. Any remaining solids were removed via centrifugation, applying the previously mentioned conditions. To facilitate precipitation, the supernatant underwent an ethanol addition at four times its volume. The ensuing precipitate, following centrifugation, was rinsed in sequence with petroleum ether, acetone, and unadulterated ethanol, The resultant was then freeze-dried to obtain crude polysaccharides, which are designated as DP. To purify the polysaccharide, the crude DP was separated on a DEAE 3epharose fast flow ion exchange column eluting with distilled water and stepwise addition of different NaCl solutions (0.2, 0.5, and 1.0 mol/L), The eluent flow rate was adjusted to 1 mL/mim, and the automatic collector collected one tube every 5 mL. Absorb each tube of sugar separately with 0.25 mL liquid, and 0.25 mL distilled water, then add 0.5 mL 5% phenol, shake well, and add 2.5 mL. The concentrated sulfuric acid was shaken and placed for 5 min, and placed in a water bath at 25 °C for 10 min. The absorption was measured at 490 nm.

#### 2.1.2. Determination of Contents of Carbohydrate and Protein

Total carbohydrate and protein content was determined referring to the methods of Liangliang Cai et al. [26] All the data were detected by a TU-1810PC instrument (Puxitongyong, Beijing, China). The polysaccharides were dissolved in 1 mg/mL solution and scanned from 200 to 400 nm with an ultraviolet spectrophotometer.

### 2.2. Animals and Grouping

24 male C57BL/6 mice, each 6 weeks old and weighing between 18–22 g, were acquired from Chengdu Dossy Experimental Animals Co., LTD (License: SCXK (Szechwan) 2020-030). These mice were housed in an SPF-grade facility with a consistent 12-h light-dark cycle. All utilized protocols adhered to the animal care guidelines of the Health Ministry of China. The animal ethics board at Sichuan University’s Huaxi Hospital approved these procedures under code 20230817003.

Polysaccharides extracted from dandelion root (abbreviated as DP) were sourced from our laboratory preparations. Post a seven-day acclimatization phase, the mice underwent a random division into four distinct groups: control, model, low-DP dosage, and high-DP dosage, with each group having 6 mice. Only the control group received normal saline during the study. For the initial five days, a 3.0% DSS solution was administered to the other three groups. From days 7 through 14, both DSS and Pd-Ib (at concentrations of 150 or 300 mg/kg/day) were delivered to the animals via gavage, depending on their group. The study concluded with the administration of 1% sodium pentobarbital (at 50 mg/kg) to the subjects, followed by euthanization through CO_2_ inhalation. Tissue samples, fecal matter, and blood were then collected. Throughout the study, meticulous records of body weight, disease activity index (abbreviated as DAI, which encapsulates changes in weight, fecal consistency, and presence of blood in the feces), and the length of the colon were maintained.

#### 2.2.1. H&E Staining Assay

After dissection, the mouse colon tissues were obtained and fixed in 4% paraformaldehyde for 24 h. Ethanol gradient dehydration, xylene transparency, and then samples were dipped in wax and embedded. The wax blocks were sliced serially with a microtome adhered to glass slides and baked and dried. Next, xylene and ethanol (H&E, C0105S, Beyotime, Beijing, China) were used to dewax the sections. Following the preparation, the tissue sections underwent staining using hematoxylin (H&E, C0105S, Beyotime, Beijing, China) for a duration of 5 min. A brief differentiation process was carried out with a 1% hydrochloric acid-alcohol solution for 3 s, followed by a quick rinse in distilled water. Subsequently, the eosin solution was employed for staining over a 1-min period, with a subsequent double rinse using distilled water. To finalize, dehydration was performed, and the samples were cleared, air-dried, and subsequently sealed utilizing neutral gum. For detailed visualization, microscopic imaging was performed.

#### 2.2.2. Transmission Electron Microscopy (TEM) Experiments

Colon samples from each mouse group underwent a fixation process using 5% glutaraldehyde, with a duration spanning 5 h. Subsequent to fixation, the samples received three separate washes using a neutral phosphate buffer, with each wash session lasting for a 10-min interval; osmium acid (0.1 mol/L) was fixed for 3 h and washed with phosphate buffer 3 times every 10 min. Then gradient dehydration was performed every 15 min at 50%, 70%, 80%, 90%, 95%, and 100% alcohol concentrations. IEC-6 cells undergo the same treatment process as colon tissue. Following the infiltration of the resin, it underwent a process of embedding. Once embedded, it was polymerized. Ultra-thin sections, with a thickness ranging between 60–80 nm, were then prepared. These sections received a dual stain using uranium and lead, subsequently air-dried under ambient conditions. The final examination of these sections was performed using a transmission electron microscope (TEM).

#### 2.2.3. TUNEL Assay

Colon tissue sections underwent a dual xylene dewaxing process, each lasting between 5–10 min. Subsequently, they were sequentially treated: initially with absolute ethanol for a duration of 5 min, followed by 90% ethanol and 70% ethanol exposures for 2 min each, concluding with a 2-min distilled water rinse. These sections were later exposed to 20 μg/mL protease K, devoid of DNase, maintaining a temperature ranging from 20 °C to 37 °C over a span of 15 to 30 min. After which, a double rinse using PBS was performed. To facilitate detection, sections were incubated with 50 μL TUNEL detection solution from AAT Bioquest, China at 37 °C for a full hour. The resulting fluorescence was detected within excitation and emission wavelength ranges of 450–500 nm and 515–565 nm, respectively, manifesting as green fluorescence.

#### 2.2.4. Western Blot Analysis

From colon tissues and IEC6 cells, total proteins were isolated using a lysis buffer. These proteins were subjected to heat at 98 °C for a short duration of 10 min. Following this, they were electrophoretically separated on a 10% SDS-PAGE and subsequently electrotransferred onto PVDF membranes. After the transfer, the membranes were blocked with a 5% non-fat milk solution for an hour at ambient temperature. Various primary antibodies were then introduced: GPX4 (1:500, A1933, abclonal, Wuhan, China), SLC7A11 (1:500, A13685, abclonal, Wuhan, China), TFR1 (1:500, A5865, abclonal, Wuhan, China), Nrf2 (1:500, A0674, abclonal, Wuhan, China), HO-1 (1:500, A19062, abclonal, Wuhan, China), and NQO-1 (1:500, A19586, abclonal, Wuhan, China), which were incubated with the membranes overnight in a 4 °C environment. This was followed by a one-hour exposure to a secondary antibody (1:7000, AS003, abclonal, Wuhan, China). For normalization, β-actin (1:5000, AC026, abclonal, Wuhan, China) served as an internal reference protein to evaluate relative protein expression levels. The membranes’ imaging was eventually carried out, with optical densities of the bands of interest being quantified using the AlphaEaseFC software4.0.0 suite from Alpha Innotech Corp, 3040 Oakmead Village Drive, Santa Clara, CA, USA.

### 2.3. Iron Ion Detection

The iron content within the colon and IEC6 cells was determined by employing the Iron Assay Kit (ml095089, Shanghai Enzyme-linked Biotechnology Co., Ltd., Shanghai, China), adhering closely to the recommended protocol by the supplier. After rinsing the tissues or cellular samples twice with chilled PBS, they were subjected to lysis in a 200 µL solution free of metal-binding agents like EDTA and citric acid. The lysed mixture was then agitated on a shaker for a duration of 2 h. A volume of 200 µL of this lysed sample was introduced into a 96-well microplate, followed by quantifying absorbance specifically at a wavelength of 593 nm.

#### 2.3.1. Immunohistochemical Experiment

Colon samples were initially preserved in a solution containing 4% neutral formaldehyde. Subsequent dehydration processes utilized a progressive series of alcohol concentrations. Once dehydrated, the samples underwent a clarity process using xylene, followed by embedding in paraffin. Upon preparing paraffin sections, they were rehydrated through dewaxing and underwent antigen retrieval, cooling down to ambient temperature. These sections were then blocked with 10% goat serum. Primary antibody MPO (1:200, Ab208670, Abcam, Cambridge, UK) was added and left undisturbed at a consistent 4 °C overnight. A biotinylated secondary antibody was then applied for a period of 30 min. For visualization, staining was carried out using 3,3′-diaminobenzidine (DAB, ZLI-9018, ZSGB, Beijing, China), succeeded by a hematoxylin counterstain. Lastly, after sealing the samples with a neutral resin, they were examined and documented utilizing a BA200 digital trinocular microscope.

#### 2.3.2. ELISA Assay

The serum was collected and tested by ELISA according to the instructions. LPS and L-DH were measured by ELISA kit mouse LPS ELISA KIT (ZC-39007), and mouse LDH ELISA KIT (ZC-38621), which were purchased and produced by Shanghai ZciBio Co., Ltd., Shanghai, China. The mouse colon tissue homogenate was collected and the working solution was made according to instructions. 4-HNE (ZC-37712), MDA (ZC-55718), and GSH (ZC-38228) were purchased and produced by Shanghai ZciBio Co., Ltd. Different groups of IEC-6 cells were collected by centrifugation to make a working solution according to the instructions to detect rat 4-HNE (ZC-35953), MDA (ZC-55718), and GSH (ZC-36690). These rat ELISA kits were purchased and produced by Shanghai ZciBio Co., Ltd. The sample was dispensed into an antibody-coated 96-well plate, acting as the reaction substrate. This was followed by a series of incubations and meticulous washing phases. Once the chromogenic substrate step concluded, Bio-Rad (Hercules, CA, USA) equipment facilitated the result readings.

#### 2.3.3. RT-qPCR Assay

RNA from the colon was extracted using Thermo Fisher’s RNA Centrifuge Column Kit (12183018A). This purified RNA was subsequently subjected to reverse transcription via the Superscript III First-Strand Synthesis System from Invitrogen. We quantified the target genes’ relative expression by employing real-time fluorescence PCR, integrating SYBR green dye (S7563, Invitrogen, Carlsbad, CA, USA) within the SYBR AmpPCR system 9700 setup. The RNA extraction procedure for IEC6 cells mirrored that of the mouse colon. Using GAPDH as the reference gene, the relative expression values for target genes were analyzed based on the 2-ΔΔCt methodology. The specific primers utilized for this research are outlined in Table 1.

#### 2.3.4. Immunofluorescence Assay

We procured colon tissue samples and prepared them for paraffin embedding. The tissues underwent a standard protocol where they were dewaxed using xylene, followed by dehydration through a series of graded ethanol solutions. Subsequently, a 0.01 mol/L citric acid buffer was applied for microwave-assisted antigen retrieval. Once the colon tissue sections were rinsed with PBS, we applied goat serum (16210064, Gibco, Grand Island, NE, USA) to block non-specific binding, allowing it to act for 10 min at ambient temperature. Subsequently, we introduced primary antibodies, specifically, Occludin (1:100, ab216327, Abcam, Cambridge, UK) and ZO-1 (1:100, GB111981, Servicebio, Wuhan, China), and left them to bind overnight in a 4 °C environment. For detection, we employed suitable secondary antibodies (A48255, Alexa Fluor 488 or 594, Invitrogen, Carlsbad, CA, USA), and after that, a 5-min room temperature incubation was carried out using 1 mg/mL DAPI (D1306, Invitrogen, Carlsbad, CA, USA) to visualize the nuclei. Visualization and imaging of these sections were performed using a Nikon Inverted fluorescence microscope (Model TE-2000U; Nikon, Tokyo, Japan), while the images were captured via SPOT digital camera (Diagnostic Instruments, Inc., Sterling Heights, MI, USA).

#### 2.3.5. Fecal Microbiota 16S rRNA Analysis

Following the guidance provided by the kit protocol, DNA was isolated from stool samples weighing close to 0.25 g utilizing the QIAamp Fast DNA Stool Mini Kit (Qiagen, Valencia, CA, USA). Subsequent PCR assays were conducted on the isolated DNA targeting the 16SV3-V4 region with the primers: forward 5′-ACTCCTACGGGAGGCAGCAG-3′ and reverse 3′-GGACTACHVGGGTWTCTAAT-5′. With the diluted genomic DNA serving as the template, we employed the KAPA HiFiHotstar ReadyMix PCR, benefiting from the kit’s high-fidelity enzyme to guarantee precision and optimal efficiency. Sequencing libraries were subsequently generated using the TruSeq Nano DNA LT kit from Illumina. The entire gut microbiota sequencing procedure was executed by Metware Company, located in Wuhan, China.

#### 2.3.6. Determination of Short-Chain Fatty Acid Content in Stool

Utilizing a methodology described in reference [20], we began by weighing 100 mg of fecal matter, combining it with double its volume in ultrapure water, and ensuring a thorough vortex homogenization under an ice bath. This mixture was then subjected to centrifugation at 12,000× *g* for a span of 10 min at 4 °C, post which the supernatant was collected. Using an Agilent gas chromatograph (GC) equipped with a silica gel capillary column (DB-Wax, J&W 30 m × 0.25 mm I.D.), the collected sample was channeled through a silica gel capillary for filtration. For our analysis, the chromatographic conditions were set with the column temperature ramping from 50 °C to 220 °C at a rate of 4 °C/min. Both the injection port and the detector were maintained at temperatures of 225 °C and 250 °C respectively. Helium acted as the carrier gas, flowing at 1.0 mL/min and adopting a split ratio of 1:20. Each standard and sample, injected in volumes of 1 μL, was analyzed for a duration of 25 min. For quantification purposes, the SCFA standard curves were generated by plotting their varying concentrations against the resultant peak areas.

#### 2.3.7. Cell Culture

The IEC-6 rat intestinal epithelial cell line, referenced as CRL21592, was sourced from the cell repository affiliated with the Chinese Academy of Medical Sciences & Peking Union Medical College. These cells were meticulously maintained in a medium of RPMI1640, enhanced with 10% fetal bovine serum. Additionally, to prevent any bacterial contamination, 100 U·mL^−1^ concentrations of both penicillin and streptomycin were added. The cell culture conditions were set at 37 °C, and a consistent 5% CO_2_ atmosphere was maintained in an incubator.

#### 2.3.8. CCK-8 Assay

IEC-6 cells in their logarithmic growth stage were carefully harvested. These cells were subsequently seeded into 96-well culture plates, with each well receiving a calculated density of approximately 5 × 103 cells. The subsequent groupings included a control, a model, and three distinct DP concentration groups: low (5 μg·mL^−1^), medium (10 μg·mL^−1^), and high (20 μg·mL^−1^). Once the cells achieved firm adhesion to the plate’s surface, the culture media was judiciously replaced with drug-infused media of varying concentrations. Following the treatment phase, the original media was suctioned off and discarded. A 10:1 mixture of fresh media and CCK-8 was prepared, of which 100 μL was dispensed into each well. This was followed by an incubation period of an hour. Absorbance measurements were then executed at 450 nm wavelength using a microplate reader from Thermo Fisher Scientific, Waltham, MA, USA.

#### 2.3.9. Apoptosis Assay

IEC-6 cell apoptosis was evaluated utilizing flow cytometric techniques. Once harvested, these cells were suspended in a solution of 500 μL binding buffer. A staining solution was prepared using 5 μL each of phycoerythrin (PE) and fluorescein 5-isothiocyanate (FITC), both sourced from Sigma-Aldrich. The cell suspension was subsequently treated with this stain for a duration of 15 min under ambient conditions. Following this, apoptosis levels and fluorescence properties of the H9c2 cells were critically examined using a BD FACSVerse flow cytometer based in the USA.

### 2.4. Statistical Analysis

For the purpose of statistical analysis, we employed the SPSS21.0 software. All results are presented as the mean complemented by the standard deviation (SD). When comparing two distinct groups, the *t*-test was utilized. In cases of more than two groups, ANOVA was the method of choice. A *p*-value less than 0.05 was the threshold for deeming differences as statistically noteworthy.

## 3. Results

### 3.1. Separation and Purification of DP

The DP was obtained by hot-water extraction, and ethanol precipitation. The resultant was then freeze-dried to obtain crude polysaccharides, which are designated as DP, the yield of DP is 16.05% (*w*/*w*).

### 3.2. Analysis of Polysaccharide and Protein Contents

The carbohydrate content of DP was 97.1% by the phenol-sulfuric acid method. and there has no protein was found in the UV scanning spectrum at 200–400 nm.

### 3.3. Protective Effect of DP on DSS-Induced UC Colitis Mice

The use of DSS to induce UC in murine subjects has gained traction in both pathological research and the formulation of therapeutic approaches, as depicted in Figure 1A. When it comes to monitoring weight changes, there was a noticeable decline in the body weight of mice from the model group when contrasted with the controls. Interestingly, by days 12 and 14, mice treated with a higher dose of DP exhibited a weight rebound compared to their counterparts in the model group (refer to Figure 1E). As shown in Figure 1B,C, DSS caused a significant shortening of colon length, while the treatment of low- and high-DP significantly inhibited the shortening of the colon length. Figure 1D illustrates the guiding relevance of the DAI score in assessing colitis severity. Notably, both the DP and control group exhibited a marked reduction in their DAI scores, pointing towards a statistically meaningful difference. Furthermore, DP demonstrates its potential in mitigating symptoms in colitis-afflicted mice, namely weight loss, consistency of feces, and fecal bleeding. When examining histopathological evidence, the model group’s colonic tissues displayed multiple maladies, such as mucosal layer degeneration and necrosis, expansion of intestinal glands, submucosal layer edema, and vascular proliferation, to name a few. By contrast, the DP high-dose group evidenced diminished colonic tissue damage. In this cohort, sporadic occurrences of degeneration and necrosis were present, but these were mainly confined to localized mucosal layers with mild inflammation. Meanwhile, in the DP low-dose group, the therapeutic outcomes seemed less optimal, with visible mucosal layer inflammation and involvement of the submucosal layer (as seen in Figure 1F).

### 3.4. Inhibitory of DP on Ferroptosis in Colonic Epithelial Cells of UC Mice

As shown in Figure 2A, the nuclei of the colon tissue cells exhibited a slight widening of the perinuclear gap, smaller mitochondria, increased membrane density, reduced and thickened mitochondrial cristae, and dilated vesicular endoplasmic reticulum on the rough surface compared with the control group. However, treatments of DP at low and high doses distinctly reversed the DSS-induced ultrastructural changes of colonic epithelial cells (Figure 2A). The results of TUNEL staining showed a large amount of green fluorescence, suggesting that DNA was damaged in the model group. The DP at low-dose and high-dose respectively caused a reduction in DSS-caused cell death in mice colon (Figure 2B,C). In addition, DSS induced an increase in iron ion content in the colonic tissue of mice. However, the iron ion content in the colon tissue of DP-fed mice was lower than that in DSS-treated mice (Figure 2D). In addition, the DSS-induce decreases in GPX4, SLC7A11, and TFR1 expression levels of colon tissue were significantly increased by DP (Figure 2E,F). The abovementioned experimental results indicated that oral ingestion of DP can alleviate DSS-induced ferroptosis of colonic cells in mice.

### 3.5. DP Enhance Antioxidant Defense in Colon of UC Mice

Ferroptosis intertwines with processes of oxidation and the dynamics of lipid metabolism. Intriguingly, despite the notable surge in colonic MPO levels prompted by DSS, both low and high dosages of DP counteracted this, as evidenced by a marked reduction in MPO expression in UC-afflicted mice (refer to Figure 3A,B). Key markers for lipid metabolism encompass 4-HNE, MDA, and GSH. A stark contrast was observed: while the model group manifested heightened 4-HNE and MDA with diminished GSH levels, DP administration painted a converse picture with lowered 4-HNE and MDA and augmented GSH levels (elucidated in Figure 3D,E). In terms of antioxidant response elements, the diminished expression of Nrf2, HO-1, and NQO-1 in the model group was evident, but DP intervention showed promising signs with elevated expressions of these markers in UC mice (highlighted in Figure 3F,G). This ensemble of data underlines DP’s capability to fine-tune the redox balance within the colonic tissues of UC mice, predominantly via the modulation of the Nrf2/HO-1/NQO-1 pathway.

### 3.6. DP Repairs Intestinal Barrier and Inhibits Inflammation in UC Mice

A compromised gut barrier serves as a distinguishing trait of UC-afflicted mice [22]. In our analysis, LPS and LDH levels in serum samples revealed intriguing findings. While the model group exhibited elevated levels of both markers relative to the control, subsequent treatment with DP, irrespective of the dosage, showcased a decline in these levels, albeit with variations in magnitude (illustrated in Figure 4A,B). Delving into the realm of immunofluorescence, DSS-induced alterations became apparent, with a notable decline in the expression of occludin and ZO-1. However, post-DP administration, a resurgence in their expression was discerned (Figure 4C,D provide insights). Transitioning to pro-inflammatory mediators in colon tissue, the model group, when juxtaposed against controls, displayed suppressed mRNA levels of IL-1β, IL-6, TNF-α, and iNOS. Yet, high-dose DP exhibited a pronounced inhibitory effect on these mediators, while its low-dose counterpart’s impact was somewhat subdued (details in Figure 4E). Culminating our findings, DP emerges as a potential therapeutic agent, adept at reinstating intestinal barrier integrity inUC.

### 3.7. DP Adjustment of Intestinal Flora

Our subsequent analysis targeted the potential influence of DP on UC via the gut microbiota. Figure 5A reveals distinct clustering patterns: the model group depicted more dispersed sample arrangements, contrasting with the tight clustering observed in the DP-treated group. This clustering points towards a similarity in the microbial community composition post-DP administration. When we shifted our focus to the genus level, there was a perceptible spatial separation between the high-dose DP samples and the model group, suggesting that DP instigates alterations in the microbial genus distribution (as visualized in Figure 5B). Moving on to overall bacterial abundance, both phylum and genus levels demonstrated a pronounced decrease in the top 10 bacteria. However, post-DP intervention, there was a noticeable resurgence in microbial richness (elaborated in Figure 5C,D). In a comparative landscape, the model group witnessed escalated abundances of *Helicobacteraceae*, *Bacteroides*, and *Lachnoclostridium*, while those of *Romboutsia*, *Lactobacillus*, and *Odoribacter* waned. Yet, DP’s therapeutic intervention marked a degree of recovery in these bacterial populations, as elucidated in Figure 5E.

### 3.8. Metabolome Study of Short-Chain Fatty Acids in Feces

Gut microbiota-derived metabolites are pivotal both in sustaining health and influencing disease progression [21]. Within the 2D visualization, marked metabolic shifts are evident in the model group when benchmarked against the control. Interestingly, the metabolic profile in UC mice following DP intervention mirrored that of the model group (as demonstrated in Figure 6A). Venturing into the 3D representation, there’s a discernible dispersion of the model group samples, while those corresponding to both low- and high-dose DP treatments exhibited tighter clustering patterns (refer to Figure 6B). It’s worth noting that, relative to controls, DSS exposure led to reduced levels of specific metabolites, namely hydroxypropionic acid, pseudolaric acid, butonate, 3-hydroxybutyrate, and valproic acid in mice. After DP treatment, these UC mice increased short-chain fatty acids. However, DSS and DP had little effect on 3-Hyisobutyric acid (Figure 6C). The above experimental results imply that DP regulation of intestinal flora may affect metabolites, especially changes in short-chain fatty acids.

### 3.9. Anti-Inflammatory and Antioxidant Effects of DP on IEC-6 Cells

To elucidate the protective role of DP on intestinal epithelial cells, we employed H_2_O_2_ to simulate oxidative damage in IEC-6 cells. Evidently, from Figure 7A, H_2_O_2_ exerted a detrimental impact on the proliferative capacity of these cells. Intriguingly, upon the introduction of various DP concentrations, we observed a dose-correlated revival of IEC-6 cell growth post-H_2_O_2_ exposure. Flow cytometry data further validated the efficacy of DP in counteracting H_2_O_2_-mediated cytotoxicity in a graded manner (represented in Figure 7B,C). Delving deeper into oxidative stress markers, 4-HNE, and MDA concentrations surged, while GSH levels dwindled in the model group relative to the controls. Yet, in comparison to the model scenario, interventions with both medium and high DP concentrations markedly ameliorated 4-HNE and MDA accumulations while elevating GSH levels (illustrated in Figure 7D–F). Turning our attention to the inflammatory milieu, elevated levels of IL-1β, IL-6, TNF-α, and iNOS were recorded in the model group vis-à-vis controls. Remarkably, high DP dosages curtailed the expression of this inflammatory quartet, with medium DP doses specifically targeting TNF-α and iNOS, as deciphered from RT-qPCR analyses (highlighted in Figure 7G). Lastly, mechanistic explorations unveiled a decline in Nrf2, HO-1, and NQO-1 expressions within the model group. In juxtaposition, DP progressively augmented the abundance of these protective proteins, as showcased in Figure 7H,I. These experimental results suggest that damage to intestinal epithelial cells coincides with experimental changes in vivo.

### 3.10. DP Inhibits Iron Death of IEC-6 Cells

Previous investigations have identified a potential role of intestinal ferroptosis in UC-afflicted mice. Consequently, we directed our focus to IEC-6 cells to understand this phenomenon further. Our observations from Figure 8A elucidate that when juxtaposed with the control set, the IEC-6 cells within the model group presented noticeable structural aberrations. There were evident deviations in nuclear regularity, with heterochromatin predominantly clustering near the nuclear periphery. Additionally, these cells exhibited mitochondrial constriction, augmented membrane density, and a discernible reduction and thickening of cristae. Intriguingly, cells under the influence of low DP concentration showed profound morphological distortions. In contrast, those treated with medium DP concentrations showed milder anomalies, and cells exposed to high DP concentrations mirrored a near-normal structural appearance. The accumulation of intracellular iron ions is one of the hallmarks of cellular ferroptosis. Progressing further, our analysis highlighted an elevation in iron ion accumulation within the model group, as opposed to the controls. However, DP application notably mitigated this accumulation, most effectively at its medium and high concentrations, as visualized in Figure 8B. Probing into proteins pivotal for ferroptosis, we discerned that GPX4, SLC7A11, and TFR1 expressions were compromised in the model group relative to the control. Notably, DP’s introduction enhanced the expression levels of these proteins, with the most pronounced effect witnessed in the high-dose category (Figure 8C,D). Cumulatively, our findings accentuate the linkage between colonic epithelial cell ferroptosis and UC.

## 4. Discussion

The animal model of DSS-induced acute colitis in mice is widely used in the study of UC, which mimics the clinical symptoms of human UC and helps to design a rational therapeutic regimen [27] In our current analysis utilizing the mouse UC model, we observed symptoms such as diminished body weight, a reduction in colon length, and an elevation in DAI scores upon DSS exposure. These manifestations closely align with findings presented by both Park [28] and Wirtz [27] in their respective investigations. However, DP supplementation significantly ameliorated the above symptoms of DSS-induced colitis. DSS-induced UC mice had severe histopathology with disruption of the mucosal layer and inflammatory infiltration of tissues. The histopathological damage was altered in UC mice after DP treatment.

Gut barrier breakdown as one of the symptoms of IBD, damage to the intestinal epithelium, and increased intestinal permeability leading to an imbalance of inflammatory cytokines are typical symptoms of UC [29]. We found that DSS-induced model mice were overloaded with iron ions and had increased colon cell death, all of which implied that iron death occurred in colon cells. Minyi Xu reported ferroptosis involves intestinal epithelial cell death in ulcerative colitis [30]. In addition, the ultrastructure of the colon tissue showed mitochondrial damage, implying that the transmission of the completed respiratory chain was blocked. In addition, Mao Li [31] reported that the blockage of the respiratory chain during iron death leads to an increase in MDA content and a decrease in GSH. Increased levels of 4-HNE are also one of the hallmarks of iron death [32]. Previous studies have shown that GPX4, SLC7A11, and TFR1 are the key proteins in ferroptosis research [33]. In vivo assays showed that DP promoted the expression of GPX4, SLC7A11, and TFR1, and also demonstrated enhanced iron transport and increased resistance to iron death in the colonic tissues of UC mice under the effect of DP.

Our observations indicate that DP’s ability to suppress intestinal epithelial cell ferroptosis might play a role in rejuvenating the intestinal barrier functionality. A notable rise in oxidative stress has been linked to diminished colonic mucosal barrier function due to a sharp decline in tight junction proteins, marking it as a pivotal factor in UC’s onset [34]. DSS’s role in distorting enzymatic antioxidant levels was counteracted by DP intervention in our study. Concurrently, we noticed an augmented expression of proteins like Nrf2, HO-1, and NQO-1. These proteins, primarily Nrf2, HO-1, and NQO-1, hold considerable significance in oxidative stress response within tissue matrices. Their expression variations can provide insights into oxidative imbalances within colonic structures [35]. As outlined by Yan Ren’s research [36], MPO expression experiences regulation in Nrf2-influenced colitis mouse models. Since neutrophils are the primary MPO-secreting cells, fluctuations in MPO activity levels within colonic mucosal tissues offer an indirect measure of neutrophil infiltration intensity and volume [37]. In our analysis, DP curtailed the heightened expression spurred by DSS. This suggests the potential efficacy of regulating interactions within Nrf2/HO-1 and Nrf2/NQO-1 routes for UC mitigation. In essence, our findings advocate that DP’s protective effect against UC likely intertwines with the Nrf2/HO-1 and Nrf2/NQO-1 signaling trajectories.

Oxidative stress, triggered by DSS, gives rise to inflammation, consequently undermining the tight junctions. This compromise allows the influx of microorganisms, amplifying the inflammatory response, and subsequently inducing ulceration [38]. In our analysis, we observed that the escalated mRNA levels of IL-1β, IL-6, TNF-α, and iNOS due to DSS exposure were effectively counteracted with high-dose DP intervention. Rapidly renewed to facilitate digestion and sustain barrier function, Intestinal epithelial cells (IEC) predominantly rely on non-inflammatory apoptosis mechanisms [39]. As UC evolves, IEC cells are strategically mobilized to compromised sites to buttress the intestinal barrier’s cohesion [40]. Nevertheless, rampant IEC mortality coupled with unchecked inflammatory reactions predominantly account for the diminished functionality of the intestinal mucosal barrier during UC episodes [41]. Elevated serum concentrations of LPS and LDH signal heightened intestinal barrier permeability [42]. ZO-1, pivotal for intestinal cohesion, is an emblematic protein indicative of the intestinal barrier’s integrity [43]. Occludin, integral for fenestration and paracellular barriers, when downregulated, can destabilize barriers across the lungs, brain, intestines, and testes, marking its association with diverse ailments [44]. In our investigations, DSS’s influence led to the diminished expression of occludin and ZO-1, which was noticeably bolstered with DP administration. Concomitantly, we identified diminished blood levels of LPS and LDH in UC-induced mice, alluding to potential barrier revitalization. Collectively, our data posits that DP enhances intestinal mucosal barrier resilience and offers a protective shield against injuries typical in UC scenarios.

Gut microbial diversity typically diminishes in UC, marked by a decline in beneficial flora and a surge in deleterious species [45]. Oral administration of DP appears to standardize the gut bacterial profiles among treated mice groups. Elevated levels of Bacteroides and Staphylococcus, recognized to negatively influence the intestinal immune system, have been documented in DSS-induced colitis mice by Wang et al. [37]. This study suggests that DP administration leads to a reduction in the aforementioned harmful bacteria, pointing towards the potential rebalancing of intestinal immunity. Additionally, Crohn’s disease has been linked to increased Helicobacteraceae levels [46,47,48]. Yet, when treated with DP, there was not only a decline in Helicobacteraceae but also a notable surge in the relative presence of Romboutsia, Lactobacillus, and Odoribacter. These species have been correlated with enhanced production of short-chain fatty acids—crucial metabolites of the gut flora. These metabolites play a pivotal role in healing intestinal injuries and inhibiting inflammatory reactions, reinforcing intestinal immune stability [48,49]. In essence, DP’s protective effects against UC may stem from its capacity to regulate specific gut microorganisms and their metabolic products.

The intestinal epithelial barrier functions as a primary shield, guarding against harmful external invaders in the gut, and it operates in response to cues from gut microbes and mucosal immunity, ensuring the host’s protection. This barrier is fundamental for gut homeostasis. UC patients often exhibit elevated colonic epithelial permeability [50,51]. This is associated with diminished concentrations of tight junction proteins between intestinal epithelial cells, which amplifies the gut’s susceptibility to microbial agents and harmful metabolites, sparking a widespread inflammatory response [52]. Delving deeper into the intestinal barrier’s role in UC, our attention was drawn to the influence of DP on IEC-6, an intestinal epithelial cell line. We employed H_2_O_2_, as suggested by prior research [53], to simulate IEC-6 damage. Concurrently, our protein analysis revealed that DP enhances the cell’s antioxidative potential and resilience against iron-induced mortality in vitro. This intervention also corrected the cellular ultrastructural anomalies. Notably, elevated DP concentrations subdued the inflammatory reactions of these epithelial cells. This aligns with the observation that DP’s impact remains consistent both in living organisms and in cell cultures. The coherence between the in vivo and in vitro findings underscores the significance of intestinal epithelial cells in the therapeutic approach of DP towards UC.

Through the oral intake of DP, there’s a noticeable increase in the fecal concentrations of acetic, propionic, and n-butyric acids in mice. It’s well understood that non-digestible carbohydrates can serve as substrates for gut microbial metabolism, leading to the generation of short-chain fatty acids [54]. Defined as saturated fatty acids with 1 to 6 carbons, SCFAs offer multiple health benefits. Their actions span from fortifying the intra-epithelial barrier, reducing its permeability, to direct engagement with immune cells. Moreover, they indirectly modulate the profiles of cytokines and chemokines. Through these combined actions, SCFAs downregulate pro-inflammatory signals and enhance immune regulatory responses [55]. This suggests that DP intake contributes to a beneficial reshaping of the gut microbiome and its metabolic output, offering potential advantages in conditions of host stress.

### Future Improvement

To purify the polysaccharide, the crude DP was separated on a DEAE sepharose fast flow ion exchange column eluting with distilled water and stepwise addition of different NaCl solutions (0.2, 0.5, and 1.0 mol/L). Four polysaccharides were separated and designated as DP1, DP2, DP3, and DP4, DP1, DP2, DP3, and DP4 accounted for 18%, 11%, 2%, and 0.5% of the total DP content, respectively. The DEAE sepharose fast flow ion exchange column chromatogram of DP is shown in Figure 9. The future study plans to explain clearly the main components of DP, the crude DP was separated on a DEAE sepharose fast flow ion exchange column eluting with distilled water and stepwise addition of different NaCl solutions (0.2, 0.5, and 1.0 mol/L). Future studies will be conducted to prove whether one or more of these four polysaccharides are responsible for the effects seen in DP.

## 5. Conclusions

The therapeutic effect and mechanism of DP to ameliorate ulcerative colitis is complex, and the therapeutic effect of DP on colitis has been clarified in this study. DP can protect the structural integrity of the intestinal epithelium by inhibiting iron death of intestinal epithelial cells, attenuating oxidative stress damage and inflammation restoring intestinal barrier homeostasis, and remodeling intestinal flora. However, the correlation between gut microbiota and metabolites still deserves further study.

At present, some products using dandelion root have been developed, but these products are relatively elementary. Through this study, the effect of DP on ulcerative colitis was further revealed, and the research basis for the further development of nutritional supplement dietary powder based on DP as a functional component was provided. The development of new products could not only increase its added value but also provide new ideas for the development of dandelion resources, which had great potential and broad prospects.

## Figures and Tables

**Figure 1 foods-12-03800-f001:**
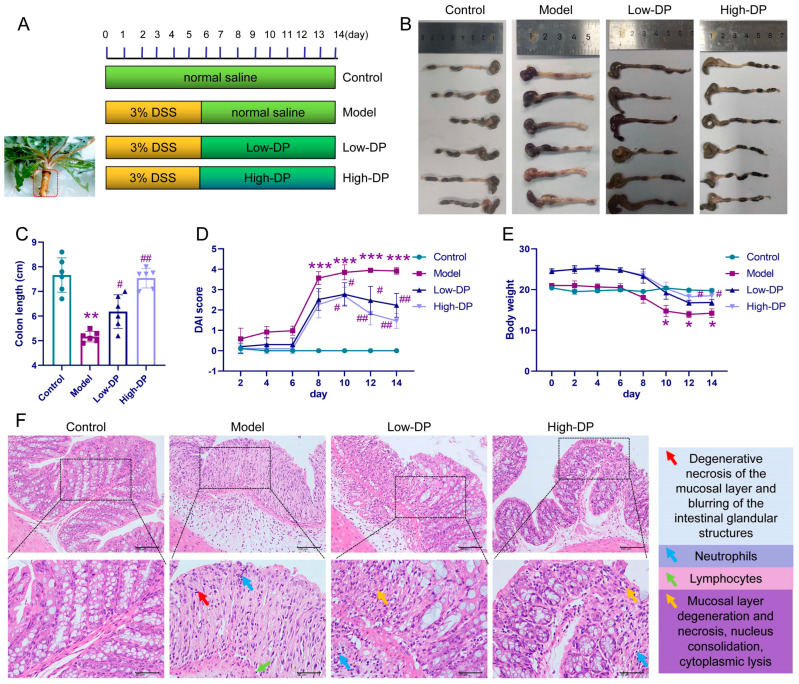
Assessment of enteritis manifestations in murine models. Panel (**A**) displays a graphical representation of the study layout. Panels (**B**,**C**) depict the colon lengths observed in mice. Panel (**D**) showcases the calculated Disease Activity Index (DAI) scores across experimental cohorts. Panel (**E**) presents the impact of dandelion root polysaccharide (DP) on murine body mass trends. Panel (**F**) reveals microscopic visuals of colon specimens post-hematoxylin-eosin (H&E) staining procedure. The computation for DAI was anchored on criteria like body mass fluctuations, fecal consistency, and observable blood in feces. Symbols * *p* < 0.05, ** *p* < 0.01, and *** *p* < 0.001 signify statistical deviations from the control set. Markers ^#^ *p* < 0.05, ^##^ *p* < 0.01 indicate significant variations from the model set.

**Figure 2 foods-12-03800-f002:**
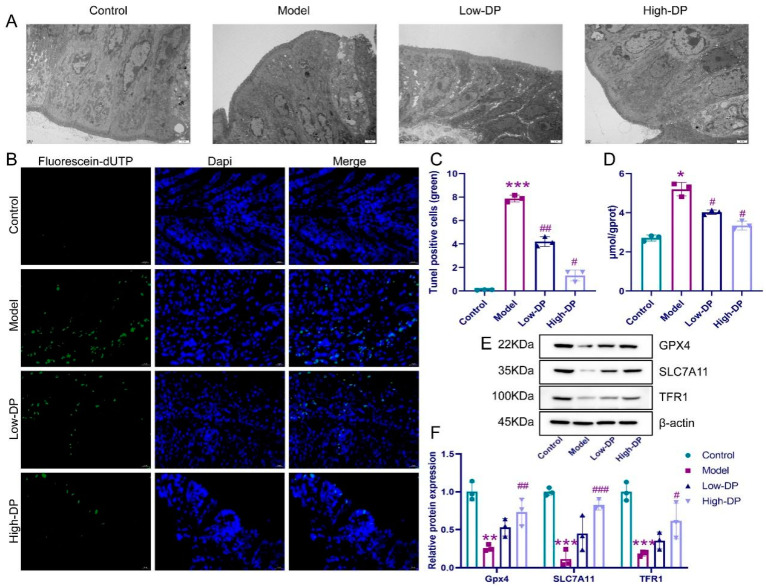
Inhibitory effect of DP on iron death in colonic epithelial cells of UC mice. (**A**): Scanning electron microscope to observe the state of Colonic tissue. (**B**,**C**): Images of colon tissue following TUNEL staining. (**D**): Determination of iron content in colonic tissue. (**E**,**F**): The relative protein levels of GPX4, SLC7A11, and TFR1. * *p* ˂ 0.05, ** *p* < 0.01 and *** *p* ˂ 0.001, compared with control group. ^#^ *p* ˂ 0.05, ^##^ *p* ˂ 0.01, and ^###^ *p* ˂ 0.001, compared with the model group.

**Figure 3 foods-12-03800-f003:**
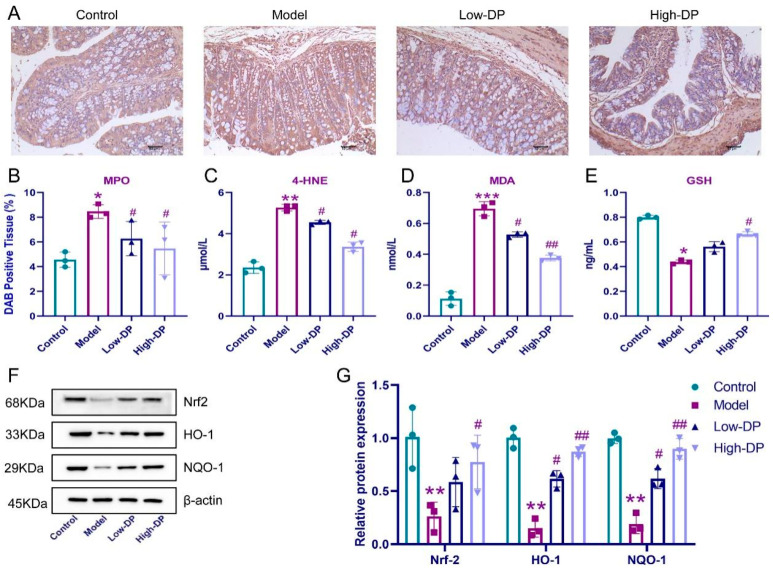
Modulation of Nrf2 anti-lipid peroxidation signaling activation by DP in the colon of UC mice. (**A**,**B**): Immunohistochemistry for MPO. (**C**–**E**): Elisa’s detection of 4-HNE, MDA, and GSH. (**F**,**G**): The relative protein levels of Nrf2, HO-1, and NQO-1. * *p* ˂ 0.05, ** *p* < 0.01 and *** *p* ˂ 0.001, compared with the control group. ^#^ *p* ˂ 0.05, ^##^ *p* ˂ 0.01, compared with the model group.

**Figure 4 foods-12-03800-f004:**
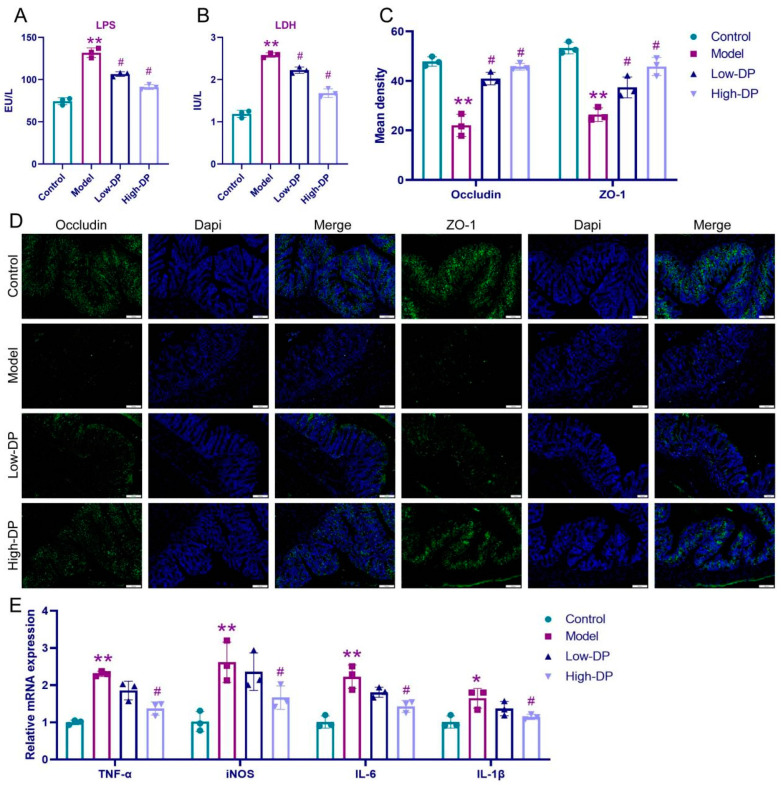
Effect of DP on the intestinal barrier as well as inflammation in UC mice. (**A**,**B**): ELISA detection of LPS and LDH content. (**C**,**D**): Immunofluorescence detection of occludin and ZO-1. (**E**): Relative mRNA of TNF-α, iNOS, IL-6 and IL-1β. * *p* ˂ 0.05 and ** *p* ˂ 0.001, compared with control group. ^#^ *p* ˂ 0.05, compared with the model group.

**Figure 5 foods-12-03800-f005:**
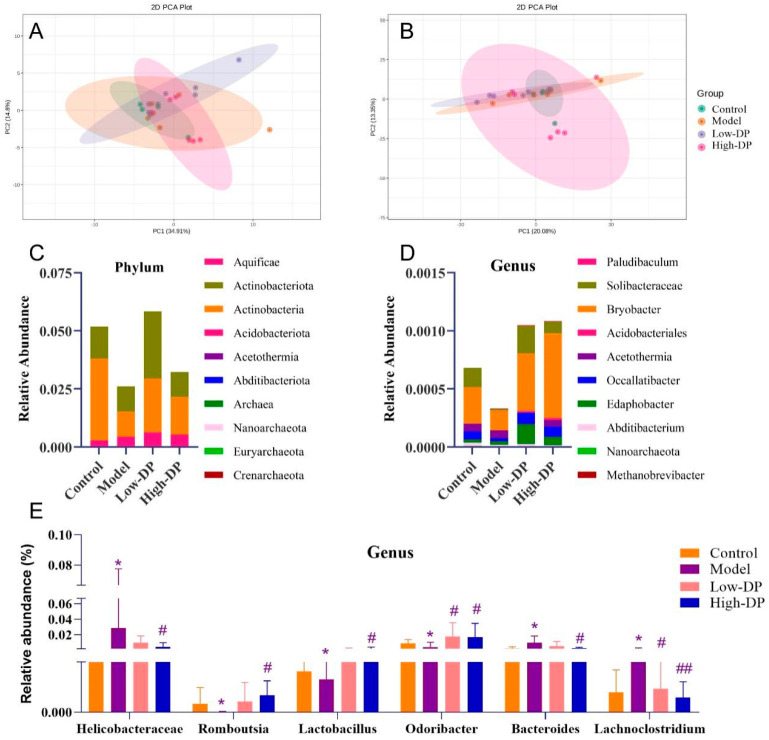
DP regulates intestinal flora regulating UC. (**A**): Beta diversity analysis sample for PCoA analysis of phylum (**A**) and genus (**B**). (**C**): The abundance of intestinal flora at the phylum level. (**D**): The abundance of intestinal flora at the genus level. (**E**): Abundance of *Helicobacteraceae*, *Romboutsia*, *Lactobacillus*, *Odoribacter*, *Bacteroides*, and *Lachnoclostridium* in each group at the genus level. * *p* ˂ 0.05, compared with control group. ^#^ *p* ˂ 0.05, ^##^ *p* < 0.01, compared with the model group.

**Figure 6 foods-12-03800-f006:**
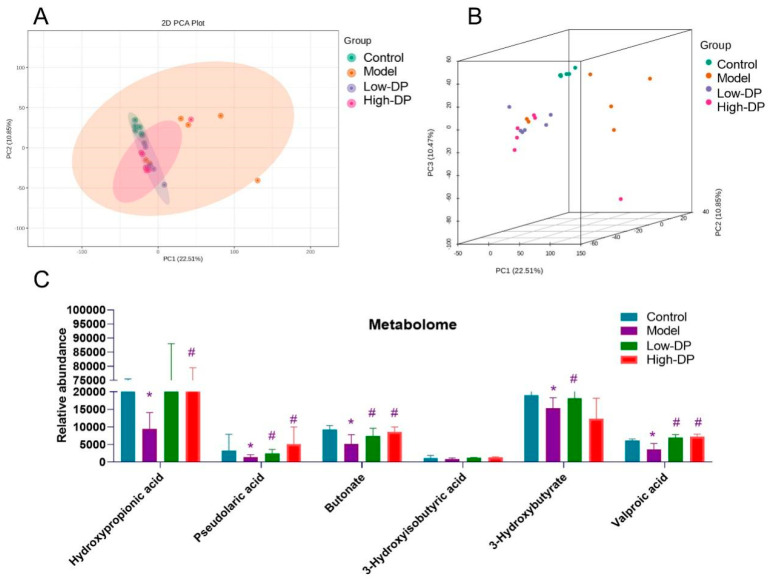
Effects of DP on short-chain fatty acids. (**A**): Metabolites 2D PCA. (**B**): Metabolites 3D PCA. (**C**): The effects of DP on short-chain fatty acids: hydroxypropionic acid, pseudolaric acid, butonate, 3-hydroxyisobutyric acid, 3-hydroxybutyrate, and valproic acid. * *p* ˂ 0.05, compared with control group. ^#^ *p* ˂ 0.05, compared with the model group.

**Figure 7 foods-12-03800-f007:**
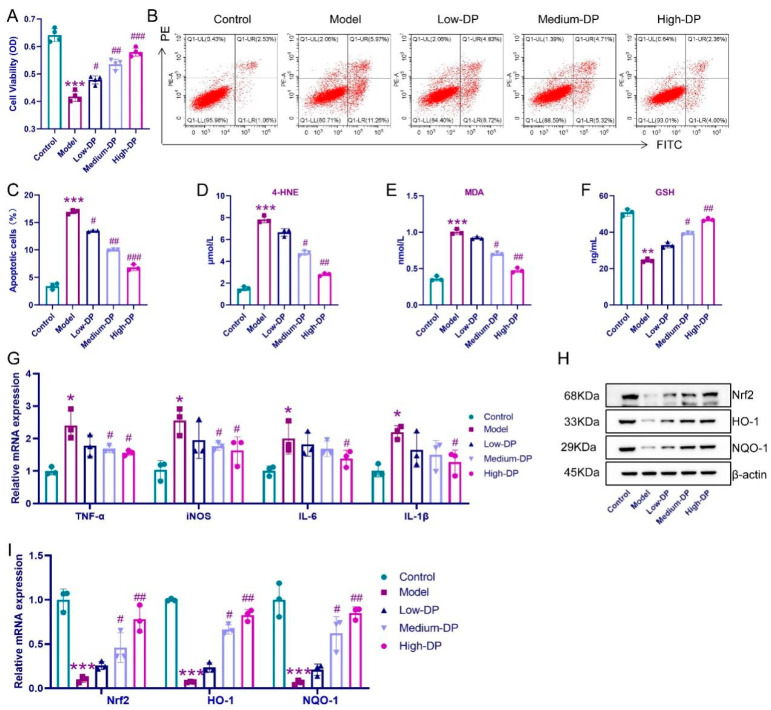
Anti-inflammatory and antioxidant effects of DP on H_2_O_2_-induced inflammation model of IEC-6 cells. (**A**): CCK-8 detects the proliferation activity of IEC-6. (**B**,**C**): Effect of DP on IEC-6 apoptosis. (**D**–**F**): Elisa’s detection of 4-HNE, MDA, and GSH. (**G**): Relative mRNA of IL-1β, IL-6, TNF-α, and iNOS. (**H**,**I**): The relative protein levels of Nrf2, HO-1, and NQO-1. * *p* ˂ 0.05, ** *p* ˂ 0.01, and *** *p* ˂ 0.0001, compared with the control group. ^#^ *p* ˂ 0.05, ^##^ *p* ˂ 0.01, and ^###^ *p* ˂ 0.0001, compared with the model group.

**Figure 8 foods-12-03800-f008:**
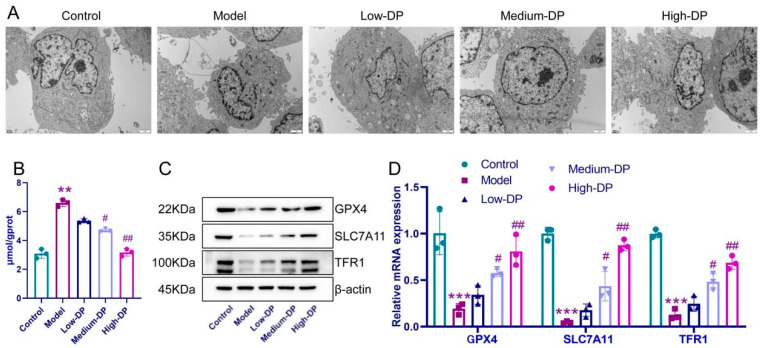
The death mechanism of IEC-6 cells induced by DP on H_2_O_2_. (**A**): Scanning electron microscope to observe the state of IEC-6 cells. (**B**): Determination of iron content in IEC-6 cells. (**C**,**D**): The relative protein levels of GPX4, SLC7A11, and TFR1. ** *p* < 0.01 and *** *p* ˂ 0.001, compared with control group. ^#^ *p* ˂ 0.05, ^##^ *p* ˂ 0.01, compared with model group.

**Figure 9 foods-12-03800-f009:**
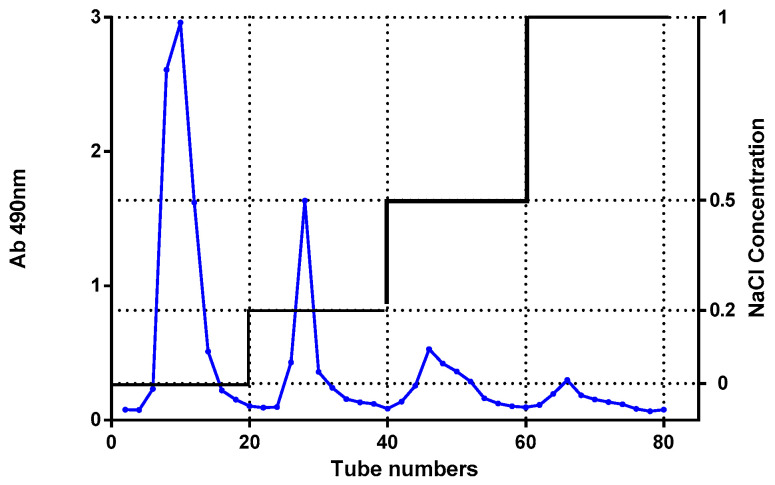
Anion exchange chromatogram of the crude DP on a DEAE sepharose fast-flow ion exchange column.

**Table 1 foods-12-03800-t001:** Primers used in this study.

Primer	Forward Primer	Reverse Primer
β-actin (mous)	CTACCTCATGAAGATCCTGACC	CACAGCTTCTCTTTGATGTCAC
IL-1β (mous)	TCGCAGCAGCACATCAACAAGAG	AGGTCCACGGGAAAGACACAGG
IL-6 (mous)	CTCCCAACAGACCTGTCTATAC	CCATTGCACAACTCTTTTCTCA
iNOS (mous)	ATCTTGGAGCGAGTTGTGGATTGTC	TAGGTGAGGGCTTGGCTGAGTG
TNF-α (mous)	ATGTCTCAGCCTCTTCTCATTC	GCTTGTCACTCGAATTTTGAGA
β-actin (rat)	GGGAAATCGTGCGTGACATT	GCGGCAGTGGCCATCTC
IL-1β (rat)	ATCCTCTCCAGTCAGGCTTCCTTGTG	AGCTCTTGTCGAGATGCTGCTGTGA
IL-6 (rat)	ACTTCCAGCCAGTTGCCTTCTTG	TGGTCTGTTGTGGGTGGTATCCTC
iNOS (rat)	AAATCCTACCAAGGTGACCTGAAAGAG	CCTGTGTTGTTGGGCTGGGAATAG
TNF-α (rat)	ATGGGCTCCCTCTCATCAGTTCC	CCTCCGCTTGGTGGTTTGCTAC

## Data Availability

The datasets used or analyzed during the current study are available from the corresponding author upon reasonable request.

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
