# Peer review of "Inhibition of IEC-6 Cell Proliferation and the Mechanism of Ulcerative Colitis in C57BL/6 Mice by Dandelion Root Polysaccharides"

_foods, 2023, doi:10.3390/foods12203800_

Round 1
Reviewer 1 Report
In this manuscript authors evaluated the molecular mechanism of dandelion root polysaccharides on ulcerative colitis in vivo and in vitro. These anti-colitis effects of dandelion root polysaccharides are assessed by various markers/parameters like H&E staining assay, TEM assay, TUNEL assay, Western blot analysis, iron ion detection, Immunohistochemical assay, ELISA assay, RT-qPCR assay, immunofluorescence assay, 16S rRNA analysis, CCK-8 assay, and apoptosis assays. At last, the author draws a conclusion that dandelion root polysaccharides could prominently inhibit the ulcerative colitis. The research question is well defined and execution is also good but there are certain queries and suggestion which authors must need to resolve.
Here are some of these:
1. I suggest revise the title to: Insight into the mechanism of dandelion root polysaccharides inhibiting ulcerative colitis in IEC-6 cells and C57BL/6 mice
2. In Figure 4 caption: Relative mRNA of IL-1β, IL-6, TNF-α, and iNOS. The order of expression needs to be consistent with the Figure 4E.
3. In Figure 6 caption: Pseudolaric acid, Butonate, 3-Hydroxyisobutyric acid, 3-Hydroxybutyrate -- The first letters of these three words need to be lowercase
Other minor errors:
Line 285 Protective Effect →Protective effect
Line 285 Colitis mice→ colitis mice
Line 378 DP Adjustment of intestinal flora→ DP adjustment of intestinal flora
Line 390 Bacterial strains require italics
Author Response
We are very sorry for our careless mistake and the relevant content has been revised :
1 the title has been changed to: Inhibition of IEC-6 cell proliferation and mechanism of ulcerative colitis in C57BL/6 mice by dandelion root polysaccharides
2 Figure 4E has been changed to: : Relative mRNA of TNF-α, iNOS, IL-6 and IL-1β.
3 In Figure 6 caption: Pseudolaric acid, Butonate, 3-Hydroxyisobutyric acid, 3-Hydroxybutyrate has been changed to: has been changed to: The effects of DP on short chain fatty acids: hydroxypropionic acid, pseudolaric acid, butonate, 3-hydroxyisobutyric acid, 3-hydroxybutyrate, and valproic acid.
4 Line 285 Protective Effect has been changed to: Protective effect
5 Line 285 Colitis mice has been changed to: colitis mice
6 Line 378 DP Adjustment of intestinal flora has been changed to: DP adjustment of intestinal flora
7 Line 390 has been changed to: Helicobacteraceae, Bacteroides, and Lachnoclostridium, while those of Romboutsia, Lactobacillus, and Odoribacter

Reviewer 2 Report
The manuscript is well-written, presents relevant results, and is consistent with the objectives. I believe that the objectives were achieved, and the discussion presented is well-founded. However, I believe that the addition of information and data is necessary for the article.
The authors must add a paragraph in the introduction that provides information about dandelion, including what it is, how it is consumed, where it is produced, its main nutraceutical components, etc.
The authors went to great lengths to demonstrate the action of dandelion polysaccharide. However, in line 89, the authors state that they obtained crude polysaccharides, which are designated as DP. In other words, the authors clearly state that they did not study a single polysaccharide but rather a mixture of polysaccharides. I don't see this as a problem, as long as the authors make it clear that they worked with a mixture of polysaccharides and not a single polysaccharide. Therefore, I request that the authors review the text with the aim of making this distinction very clear in the text.
As DP is not a pure polysaccharide but a mixture, the authors must chemically characterize DP, demonstrating at least the percentage of polysaccharide present in DP.
The authors must also demonstrate whether DP is contaminated with proteins, phenolic compounds, or other molecules that generally contaminate polysaccharides and may have pharmacological effects. This will ensure that the effects observed by the authors result from the polysaccharide and not from other molecules.
In the methods section, the authors forgot to describe how they evaluated DP as an inhibitor of iron-induced cell death in IEC-6 cells."
the text is well written, I didn’t identify any errors
Author Response
As suggested by the reviewer, we have added more references in the introduction part and the part of Separation and purification of DP. Please see the attachment for the specific modification.

Reviewer 3 Report
The authors evaluated the potential therapeutic benefits of dandelion poly-7 saccharide (DP) in a murine model of ulcerative colitis (UC) caused by 3% dextran sodium sulfate (DSS). The findings demonstrated that DP was effective in reducing weight loss, decreasing disease activity index scores, normalizing colon length, and treating histological abnormalities in afflicted mice. DP restored mitochondrial damage in colonic cells by increasing iron transport and inhibiting iron death. Furthermore, DP boosted antioxidant capability as measured by increased expression of Nrf2, HO-1, NQO-1, and GSH and decreased levels of MDA and 4-HNE. Concurrently, DP inhibited inflammatory mediators such as IL-1, IL-6, and TNF-, while increasing the expression of occludin and ZO-1. Analysis of metabolites indicated that DP counteracted metabolic disturbances and augmented the levels of short-chain fatty acids in UC mice, being able to modulate the gut microbiome, rectify metabolic imbalances, and bolster the intestinal barrier as a therapeutic approach to UC.
In conclusion, the mentioned manuscript has not only fundamental, but also translational significance, with analysed parameters that support the obtained results in terms of both quality and quantity.
Author Response
Thanks for reviewer‘s comments,we appreciate it very much for this good suggestion, and we have done it according to your ideas. thank you very much.
Round 2
Reviewer 2 Report
The authors modified the title to indicate that the study refers to polysaccharides and not a single polysaccharide.
The authors added information about dandelion root in the introduction, as requested.
The authors cannot simply assume that the crude extract is composed of only polysaccharides. It is extremely important that the authors indicate what the main components of the crude extract are. They must demonstrate the content of carbohydrates, proteins, phenolic compounds, etc. of the crude extract.
The authors present a fractionation of the crude layer of dandelion root polysaccharides using ion exchange chromatography. However, they do not provide any further data on these polysaccharides, neither physicochemical nor physiological/pharmacological data. Therefore, I believe that the position in which this data was presented is incorrect.
I suggest that the authors present this data as the last figure of the article and inform readers that this fractionation was done to show the main components of the crude extract (DP) and that future studies will be carried out to demonstrate whether one or more of these four polysaccharides are responsible for the effects seen with DP.
In the methods section, it is crucial that the authors indicate how they tracked the elution of material from the column. Dubois method? This is not clear.
It is also crucial that the authors confirm the glycidic nature of each of the four peaks obtained from chromatography.
In the title of figure nine, it is stated: "Figure 9. Effect of DP on H2O2-induced iron death in IEC-6 cells." However, I could not find any information in the text on how H2O2-induced iron death was performed. It is crucial that the authors add this information to the text.
In line 334 The authors use the word “deproteinization”. However, in methods they did not use deproteinization. What is the correct?
The English is OK
Author Response
1 The authors cannot simply assume that the crude extract is composed of only polysaccharides. It is extremely important that the authors indicate what the main components of the crude extract are. They must demonstrate the content of carbohydrates, proteins, phenolic compounds, etc. of the crude extract.
Reply: Dear editor, thank you very much for your valuable suggestions. The specific content has been revised in the article.
2 The authors present a fractionation of the crude layer of dandelion root polysaccharides using ion exchange chromatography. However, they do not provide any further data on these polysaccharides, neither physicochemical nor physiological/pharmacological data. Therefore, I believe that the position in which this data was presented is incorrect.
Reply: Dear editor, thank you very much for your valuable suggestions. We appreciate it very much for this good suggestion, and we have done it according to your ideas, and present this data as the last figure of the article.
3 I suggest that the authors present this data as the last figure of the article and inform readers that this fractionation was done to show the main components of the crude extract (DP) and that future studies will be carried out to demonstrate whether one or more of these four polysaccharides are responsible for the effects seen with DP.
Reply: Dear editor, thank you very much for your valuable suggestions. We appreciate it very much for this good suggestion, and we have done it according to your ideas, and present this data as the last figure of the article.
4 In the methods section, it is crucial that the authors indicate how they tracked the elution of material from the column. Dubois method? This is not clear.
Reply: Dear editor, thank you very much for your valuable suggestions. how to track the elution of material from the column,we used phenol-sulfuric acid method was used for trace detection. Enzyme-labeled instrument was used at 490nm.
5 It is also crucial that the authors confirm the glycidic nature of each of the four peaks obtained from chromatography.
Reply: We thank the reviewer for the interesting comment, which will help us to further improve our study in the future. Now we haven't really thought about it in detail yet. Again, we thank the reviewer and, in future study, we will certainly consider challenging this interesting hypothesis.
6 In the title of figure nine, it is stated: "Figure 9. Effect of DP on H2O2-induced iron death in IEC-6 cells." However, I could not find any information in the text on how H2O2-induced iron death was performed. It is crucial that the authors add this information to the text.
Reply: Dear editor, thank you very much for your valuable suggestions. In fact, figure 8 and figure 9 are combined to explain the effect of DP on ferroptosis in IEC-6 cells. In order to avoid misunderstanding of readers, I have supplemented the figure 9 legend title and result content. The accumulation of intracellular iron ions is an important sign of cell ferroptosis, as we can learn from figure 9B. In addition, I am willing to actively cooperate and answer any questions about figure 9.
7 In line 334 The authors use the word “deproteinization”. However, in methods they did not use deproteinization. What is the correct?
Reply: Dear editor, thank you very much for your valuable suggestions. We are very sorry for our incorrect writing “deproteinization”and it is rectified at Line 334.
